# Dual functions of a small regulatory subunit in the mitochondrial calcium uniporter complex

**Ming-Feng Tsai[1,2†], Charles B Phillips[1,2†], Matthew Ranaghan[1], Chen-Wei Tsai[1,2], Yujiao Wu[1,2], Carole Williams[1,2], Christopher Miller[1,2*]**

[1]Department of Biochemistry, Brandeis University, Waltham, United States;
[2]Howard Hughes Medical Institute, Brandeis University, Waltham, United States

**Abstract** Mitochondrial $Ca^{2+}$ uptake, a process crucial for bioenergetics and $Ca^{2+}$ signaling, is catalyzed by the mitochondrial calcium uniporter. The uniporter is a multi-subunit $Ca^{2+}$-activated $Ca^{2+}$ channel, with the $Ca^{2+}$ pore formed by the MCU protein and $Ca^{2+}$-dependent activation mediated by MICU subunits. Recently, a mitochondrial inner membrane protein EMRE was identified as a uniporter subunit absolutely required for $Ca^{2+}$ permeation. However, the molecular mechanism and regulatory purpose of EMRE remain largely unexplored. Here, we determine the transmembrane orientation of EMRE, and show that its known MCU-activating function is mediated by the interaction of transmembrane helices from both proteins. We also reveal a second function of EMRE: to maintain tight MICU regulation of the MCU pore, a role that requires EMRE to bind MICU1 using its conserved C-terminal polyaspartate tail. This dual functionality of EMRE ensures that all transport-competent uniporters are tightly regulated, responding appropriately to a dynamic intracellular $Ca^{2+}$ landscape.

***For correspondence:** cmiller@brandeis.edu

[†]These authors contributed equally to this work

**Competing interests:** The authors declare that no competing interests exist.

## Introduction

$Ca^{2+}$ regulation of key mitochondrial processes such as ATP production and initiation of apoptosis is controlled by precise balance of $Ca^{2+}$ influx and efflux across the mitochondrial inner membrane (*Gunter et al., 2000*; *Rizzuto et al., 2012*). Studies in the 1960s and '70s established that mitochondria from most eukaryotes, except for certain yeast species, can take up large quantities of $Ca^{2+}$ from the cytosol into the matrix through a mechanism that is membrane potential dependent and strongly inhibited by ruthenium compounds such as Ru360 (*Carafoli and Lehninger, 1971*; *Deluca and Engstrom, 1961*; *Ying et al., 1991*). A few years ago, the field witnessed a groundbreaking achievement — identification of the *MCU* gene (*Baughman et al., 2011*; *De Stefani et al., 2011*). The 35-kDa MCU protein oligomerizes with unknown stoichiometry to form a $Ca^{2+}$-selective pore (*Baughman et al., 2011*). MCU possesses two transmembrane helices (TMHs) connected by a short loop that hosts a signature sequence (DIME) thought to contribute to a $Ca^{2+}$-selective permeation site. The N- and C-terminal regions of MCU are exposed to the mitochondrial matrix, each with a coiled-coil sequence of unknown function.

It was subsequently found that MCU forms a complex with the mitochondrial $Ca^{2+}$ uptake protein 1 (MICU1), which has co-evolved with MCU since early eukaryotic evolution (*Baughman et al., 2011*; *Bick et al., 2012*). In humans, MICU1 has two additional homologues, MICU2 and the neuron-specific MICU3 (*Plovanich et al., 2013*). The MICUs serve as the $Ca^{2+}$-sensing gate that confers $Ca^{2+}$-dependence to opening of the $Ca^{2+}$-selective pore (*Csordas et al., 2013*; *Mallilankaraman et al., 2012*). In resting cellular conditions, where cytoplasmic $Ca^{2+}$ is low, MICUs shut the pore to prevent excessive $Ca^{2+}$ influx into the matrix, a dangerous process that could diminish inner membrane

**eLife digest** Like all power plants, mitochondria – the compartments inside our cells that supply energy – must adjust their energy output to match fluctuations in demand. Inside cells, the levels of calcium ions in the cytoplasm often signal such demands. Mitochondria therefore control their calcium ion levels with tightly regulated, membrane-embedded proteins that move calcium ions into and out of the mitochondria. One of these membrane machines, the mitochondrial calcium uniporter (MCU) complex, is a "smart channel" that admits calcium ions into the mitochondria only when their cytoplasmic levels exceed a threshold.

The MCU complex contains four essential proteins: MCU, which forms the pore through which the calcium ions enter the mitochondrion; MICU1 and MICU2, which act as "gatekeepers", opening the pore only when the cell contains high levels of calcium ions; and EMRE, a small, mysterious protein. Why is EMRE required for the channel's operation, and how does it fit into the four-protein complex?

By comparing EMRE proteins from different species, constructing mutant forms of EMRE, and recording calcium ion transport in mitochondria from cultured human cells, Tsai, Phillips et al. show that EMRE has two key roles. First, it snuggles up against the MCU protein and forms an essential part of the calcium ion-selective pore. Second, it acts as molecular glue to fix the calcium ion-sensing MICU gatekeepers to the pore. These two linked functions ensure that the MCU complex switches on only when the cell contains high levels of calcium ions, preventing the cell becoming catastrophically overloaded with calcium ions and cell death.

Challenges for the future are to purify the MCU complex and reconstitute its ability to transport calcium ions from its component parts. This will help to determine the structure of the channel.

potential and trigger apoptotic cell death. Transient elevation of $Ca^{2+}$ to the low μM range, detected by EF-hands in MICUs, releases this inhibition to open the channel (*Csordas et al., 2013*; *Kamer and Mootha, 2014*). To avoid confusion on nomenclature, we henceforth refer to the Ru-360 sensitive mitochondrial $Ca^{2+}$ channel complex as the 'uniporter complex,' a molecular assembly of the pore-forming MCU protein along with associated regulatory subunits.

Recently, using quantitative mass spectroscopy, Mootha and colleagues discovered yet another component of the uniporter complex: the essential MCU regulator (EMRE), a small (~10 kDa) inner membrane protein found only in metazoa (*Sancak et al., 2013*). EMRE possesses a single TMH and a highly conserved C-terminal polyaspartate tail, typically composed of one glutamate followed by 5–7 aspartates. In humans, MCU-EMRE interaction is absolutely required for $Ca^{2+}$ permeation via MCU (*Kovacs-Bogdan et al., 2014*; *Sancak et al., 2013*). However, an MCU homologue in *D. discoideum,* a species belonging to the EMRE-lacking Amoebazoa group in protists, is fully capable of conducting $Ca^{2+}$ (*Kovacs-Bogdan et al., 2014*). The question naturally arises: what might be the physiological importance for MCU to become strictly dependent on EMRE for function in humans? What would be the consequence if human MCU could transport $Ca^{2+}$ without EMRE?

We address these questions by mounting an extensive investigation of EMRE. We first sought to determine the protein's transmembrane topology, a problem that cannot be definitively resolved by standard protease digestion assays (*Baughman et al., 2011*; *Vais et al., 2016*) due to the small size of the protein's extra-membrane regions. Two alternative strategies – directed mass-tagging and MCU-EMRE fusion construction - establish that EMRE exposes its N-terminal region to the matrix and C-terminus to the intermembrane space (IMS). Mutagenesis screening and domain-interaction analysis further demonstrate that EMRE supports $Ca^{2+}$ transport by using its TMH to bind to MCU through its first TMH (TMH1). Moreover, EMRE also interacts with MICU1 via its C-terminal polyaspartate tail, a molecular contact that turns out to be crucial to retain MICUs in the uniporter complex to gate the MCU pore. These results lead to a molecular model wherein the dual 'MCU-activating' and 'MICU-retaining' functionalities of EMRE together play a crucial role in orchestrating uniporter responses to intracellular $Ca^{2+}$ signaling.

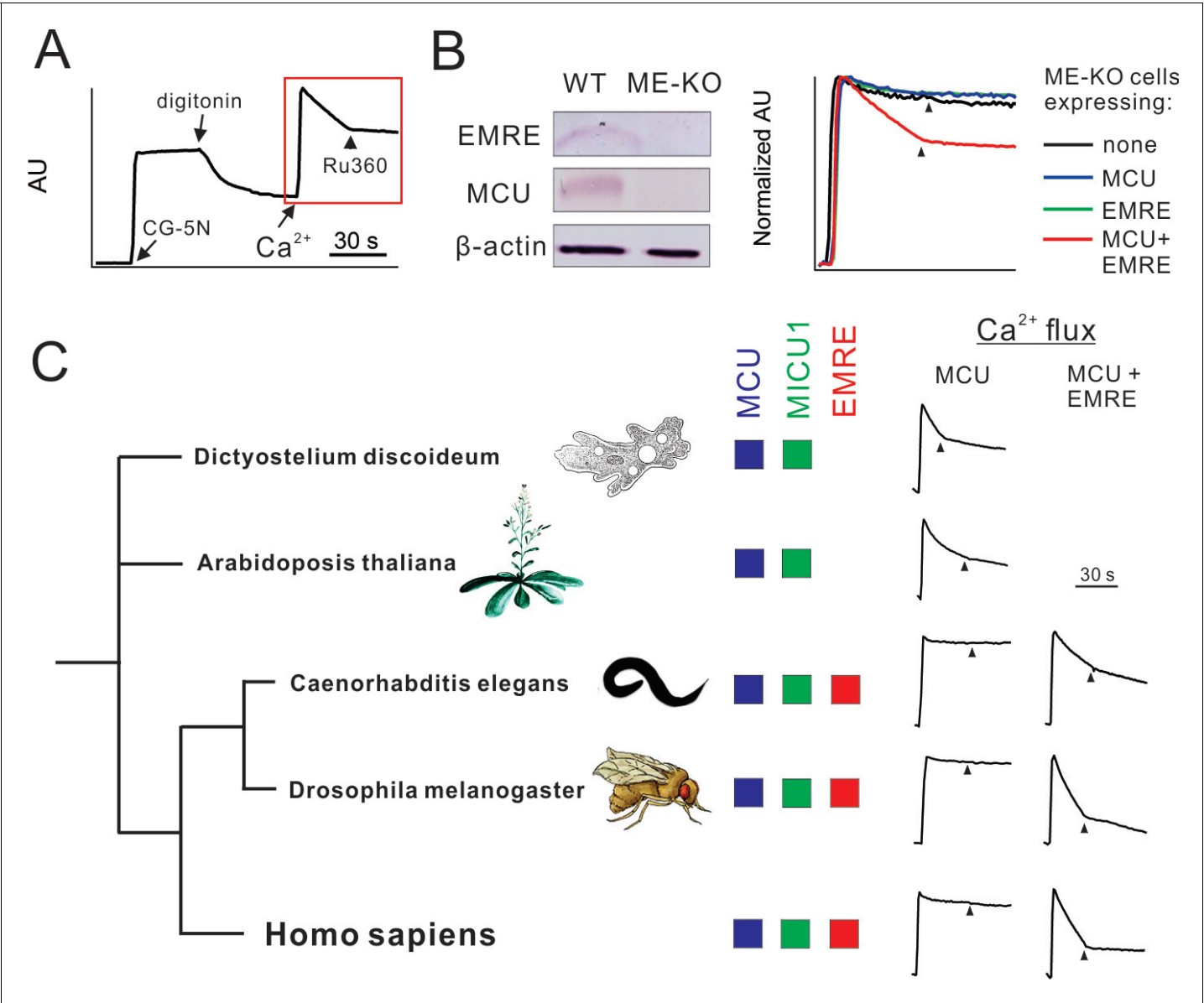

**Figure 1.** Functional analysis of uniporter in various species. (**A**) A representative fluorescence-based Ca$^{2+}$ flux experiment. (**B**) Characterization of ME-KO HEK 293 cells. *Left*: western analysis comparing EMRE, MCU, and actin expression in WT or ME-KO cells. *Right*: Loss of MCU-mediated Ca$^{2+}$ uptake in ME-KO cells, and rescue by delivering both MCU and EMRE genes. (**C**) Activity of uniporters in species indicated. Ca$^{2+}$ flux experiments were performed using ME-KO cells expressing MCU alone or MCU and EMRE from the same species.

## Results

### Functional dependence of MCU on EMRE in various species

To study uniporter subunits without interference from native mitochondrial proteins, we employed CRISPR/Cas9 to produce MCU-knockout (KO), EMRE-KO, or MCU/EMRE double KO (ME-KO) HEK 293 cell lines. A standard Ca$^{2+}$ flux assay was used to evaluate uniporter activity. In a typical experiment, HEK cells were permeabilized with digitonin in the presence of a Ca$^{2+}$-sensing fluorophore (CG-5N) and then treated with 10 μM extracellular Ca$^{2+}$ (*Figure 1A*). In WT cells, Ca$^{2+}$ is rapidly sequestered by mitochondria, and Ru360, a potent MCU inhibitor, arrests the uptake immediately (*Figure 1A*). (Henceforth for clearer data presentation, only the response of permeabilized cells to Ca$^{2+}$ will be presented, as in the red box in *Figure 1A*, with arrowheads indicating Ru360 addition.)

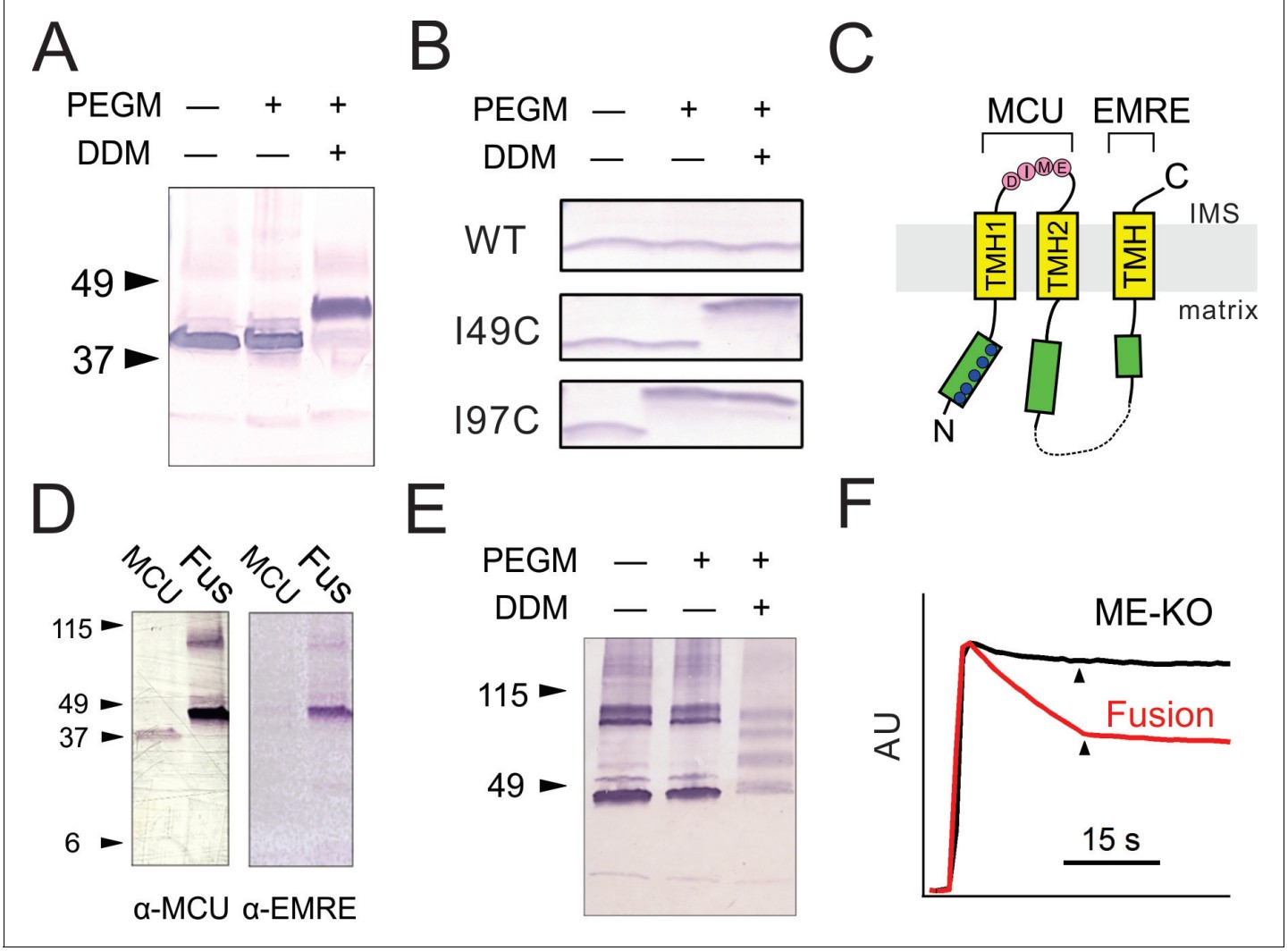

**Figure 2.** Transmembrane orientation of MCU and EMRE. (A) Western blot analysis of WT-MCU response to PEGM, in the absence or presence of DDM detergent, with molecular weight marker positions indicated on left. (B) PEGM treatment of WT, I49C, or I97C EMRE. EMRE's molecular weight is ~10 kDa. (C) Cartoon illustrating the proposed membrane orientation of MCU and EMRE. The N-terminus of EMRE is fused to the C-terminus of MCU (dashed line). Blue circles: native cysteines. Green boxes: soluble region. (D) WT-MCU or MCU-EMRE fusion protein (Fus) probed with anti-MCU (left) or anti-EMRE (right) antibodies. (E) MCU-EMRE fusion treated with PEGM. In the presence of DDM, PEGM treatment produces 4 bands, representing fusion proteins with various numbers of cysteines modified. (F) Mitochondrial $Ca^{2+}$ uptake in ME-KO cells with (red) or without (black) expression of the MCU-EMRE fusion protein. See also *Figure 2—figure supplement 1*.

The following figure supplement is available for figure 2:

**Figure supplement 1.** Uniporter function supported by indicated EMRE mutants.

Consistent with previous reports (*Baughman et al., 2011*; *De Stefani et al., 2011*; *Sancak et al., 2013*), EMRE-, MCU-, or ME-KO mitochondria are completely devoid of uniporter activity, a deficiency rescued by supplying the deleted genes (*Figure 1B*).

Genome sequences imply that MCU and MICU proteins are ancient in eukaryotic evolution, while EMRE sequences appear only among metazoa. Indeed, MCU homologues from an amoeba, *D. discoideum*, and a green plant, *A. thaliana*, both of which lack EMRE, can alone mediate rapid, Ru360-sensitive mitochondrial $Ca^{2+}$ uptake in ME-KO cells (*Figure 1C*). In contrast, metazoan MCU homologues from *C. elegans* and *D. melanogaster* require EMRE to transport $Ca^{2+}$ (*Figure 1C*), as in

humans. This striking difference raises questions regarding the biological purpose of EMRE in animals and offers opportunities for attacking questions of molecular mechanism.

## Transmembrane orientation of EMRE

To approach the physiological importance of EMRE's regulatory function, we first ask how it assembles with other uniporter subunits into a channel complex. The orientation of EMRE in the inner membrane was determined by a cysteine-modification, mass-tagging method. In a typical experiment, mitoplasts (outside-out submitochondrial vesicles lacking the outer membrane) prepared from HEK 293 cells are incubated with a 5-kDa, membrane impermeant thiol-reactive reagent, polyethylene glycol maleimide (PEGM). If the protein of interest has a cysteine exposed to the external solution, PEGM would react with this cysteine and thus increase the protein's molecular weight.

We first validated the assay on the known orientation of MCU (*Kamer and Mootha, 2015*; *Murgia and Rizzuto, 2015*). Human MCU possesses five cysteines, all in the N-terminal domain. If this region faces outward towards the IMS, PEGM would increase MCU's mass by 5 kDa per residue modified. Experiments (*Figure 2A*), however, show that MCU mobility on SDS-PAGE is not altered by PEGM treatment unless the mitoplast membrane is first disrupted by the mild detergent dodecyl maltoside (DDM). The results thus confirm MCU's $N_{in}$-$C_{in}$ orientation, with both the N- and C-termini residing in the matrix.

To determine EMRE topology, we constructed mutants with unique cysteines (I49C or I97C) engineered on either side of the TMH of the naturally cysteineless EMRE. Like WT, these mutants support MCU-dependent $Ca^{2+}$ flux in EMRE-KO cells (*Figure 2—figure supplement 1*). In intact

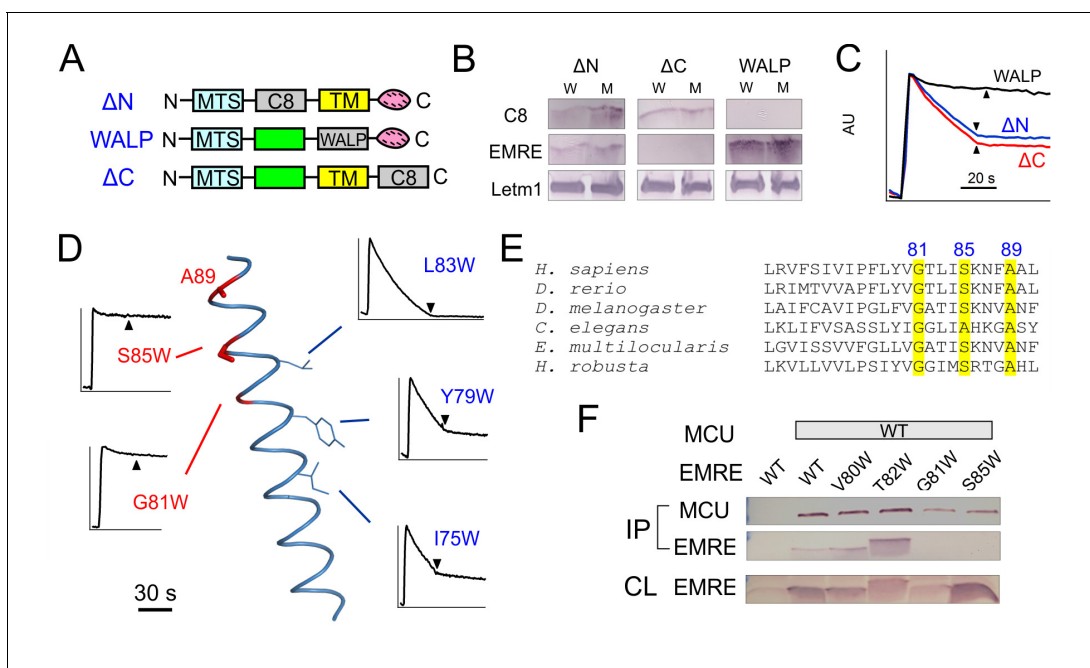

**Figure 3.** MCU-interacting residues in EMRE. (**A**) EMRE constructs with indicated regions substituted by either the C8 epitope or the WALP helix. MTS: mitochondrial targeting sequence. Green boxes: soluble regions. Pink ovals: polyaspartate tail. (**B**) The presence of these mutants in whole cell lysate (W) or isolated mitochondria (M). ΔC-EMRE is not detectable by the EMRE antibody because the C-terminal truncation removes the epitope. (**C**) Mitochondrial $Ca^{2+}$ uptake in EMRE-KO cells expressing WALP-, ΔN-, or ΔC-EMRE. (**D**) Diagram summarizing Trp scan of the EMRE TM helix. Red shows positions where Trp substitution reduces the rate of $Ca^{2+}$ uptake by >50%. (**E**) Sequence alignment of EMRE TM helix. Yellow indicates residues intolerant to Trp substitutions in human EMRE. (**F**) Co-IP experiments using 1D4-tagged MCU immobilized in 1D4 affinity columns to pull down indicated EMRE mutants. IP: elution, analyzed using indicated antibody. CL: whole cell lysate input. Upper panel: proteins being expressed in ME-KO cells. Leftmost lane: MCU-free control to rule out non-specific binding of EMRE in the 1D4 column. See also *Figure 3—figure supplement 1*.

The following figure supplement is available for figure 3:

**Figure supplement 1.** Functional impact of mutations in EMRE's TM helix.

mitoplasts free of native EMRE protein, PEGM readily labels I97C-EMRE but fails to react with I49C or the cysteineless WT protein, while after detergent pretreatment both mutants are labeled (*Figure 2B*). EMRE thus adopts a $N_{in}$-$C_{out}$ orientation, with its C-terminal tail facing the IMS.

*Figure 2C* summarizes the inner-membrane topology of MCU and EMRE inferred here. This was further verified by fusing EMRE onto the C-terminus of MCU, thus forcing the orientation of the two proteins in tandem to conform to the above molecular picture (*Figure 2C*). The fusion construct was tested in ME-KO cells, where it was expressed as a full-length protein detectable by both MCU and EMRE antibodies (*Figure 2D*). As with WT MCU, PEGM fails to modify any of the five native cysteines in the N-terminus without detergent pretreatment (*Figure 2E*), thus implying that the fusion protein is inserted homogeneously into the inner membrane in a proper $N_{in}$-$C_{out}$ orientation. Moreover, the MCU-EMRE fusion protein mediates robust, Ru360-sensitive mitochondrial $Ca^{2+}$ uptake (*Figure 2F*), a powerful result corroborating the transmembrane orientation cartooned in *Figure 2C*.

## Mapping the EMRE transmembrane helix

We next investigate how EMRE interacts with MCU to support $Ca^{2+}$ permeation. This issue is addressed using three EMRE variants (*Figure 3A*), with N- or C-termini largely deleted by replacing it with a foreign 'C8' epitope (PRGPDRPEGIEE) (*Abacioglu et al., 1994*) into either region, or with the TMH substituted by an artificial transmembrane 'WALP' helix (GWWLALALALALALALWWA) (*Killian et al., 1996*). These mutants, named ΔN-, ΔC-, or WALP-EMRE, are all properly targeted to EMRE-KO mitochondria (*Figure 3B*). Both ΔN- and ΔC-EMRE fully support uniporter function, but cells expressing WALP-EMRE exhibit no uniporter activity (*Figure 3C*). The results are surprising, as the strict conservation of EMRE's C-terminal polyaspartate tail implies that it should carry out some sort of crucial function.

To locate EMRE residues critical for MCU-mediated $Ca^{2+}$ transport, we performed tryptophan scanning mutagenesis to cover the entire transmembrane region (S64 to A92). This classical strategy posits that the large tryptophan side chain introduced at a protein interface is more likely to disrupt helical packing and hence function than when projecting towards lipid (*Hong and Miller, 2000*; *Sharp et al., 1995*). The results (*Figure 3D*, *Figure 3—figure supplement 1*) highlight a rather clean segregation of Trp-sensitive vs insensitive positions on a helical-wheel diagram. In particular, Trp substitution at G81 or S85 completely eliminates $Ca^{2+}$ uptake via the uniporter complex. These two residues belong to a conserved Gxxx[G/A/S] motif (*Figure 3E*), frequently found to mediate packing of TMHs in membrane protein structures (*Russ and Engelman, 2000*).

Co-immunoprecipitation (co-IP) experiments further confirm the GxxxS sequence as crucial to MCU-EMRE complex formation. EMRE variants were co-expressed with MCU carrying a C-terminal '1D4' epitope (TETSQVAPA) (*MacKenzie et al., 1984*) in ME-KO cells, and the MCU-EMRE complex was immobilized on a 1D4 affinity column for downstream analysis. WT EMRE is captured by MCU, but disruption of the GxxxS region by the G81W or S85W mutation prevents this association, while Trp substitution of residues elsewhere on the TMH does not (*Figure 3F*). These results taken together show that EMRE binds to MCU using a Gxxx[G/A/S] motif in the C-terminal half of its TMH, and that this binding is necessary to activate the $Ca^{2+}$-conducting pore in the uniporter complex.

## MCU recognition of EMRE

To examine the MCU side of the interaction with EMRE, we launched Trp-perturbation mutagenesis of both TMHs (residues L234 – T254 in TMH1; Y268 – V283 in TMH2). All mutants were expressed to near WT levels in MCU-KO cells (*Figure 4—figure supplement 1*), and were classified as either low- or high-impact on $Ca^{2+}$ transport function (*Figure 4A*, *Figure 4—figure supplement 1*). Of the 16 Trp mutations in TMH2, only two (F269W, T271W, located near the N-terminal end of the helix) induce severe functional defects, as if most TMH2 residues project either to the lipid bilayer or an aqueous cavity. In contrast, the six Trp-sensitive positions in TMH1 tend toward one side of a helical wheel diagram, suggesting that this high-impact face might pack against other TMHs in the MCU-EMRE complex.

If a Trp mutation in MCU perturbs the function of human uniporter solely by interrupting EMRE binding, it should cause negligible functional impact when introduced at corresponding positions in *D. discoideum* MCU, because this homologue transports $Ca^{2+}$ without EMRE (*Figure 1C*). We therefore introduced each of the eight high-impact Trp substitutions (*Figure 4A*) into *D. discoideum* MCU

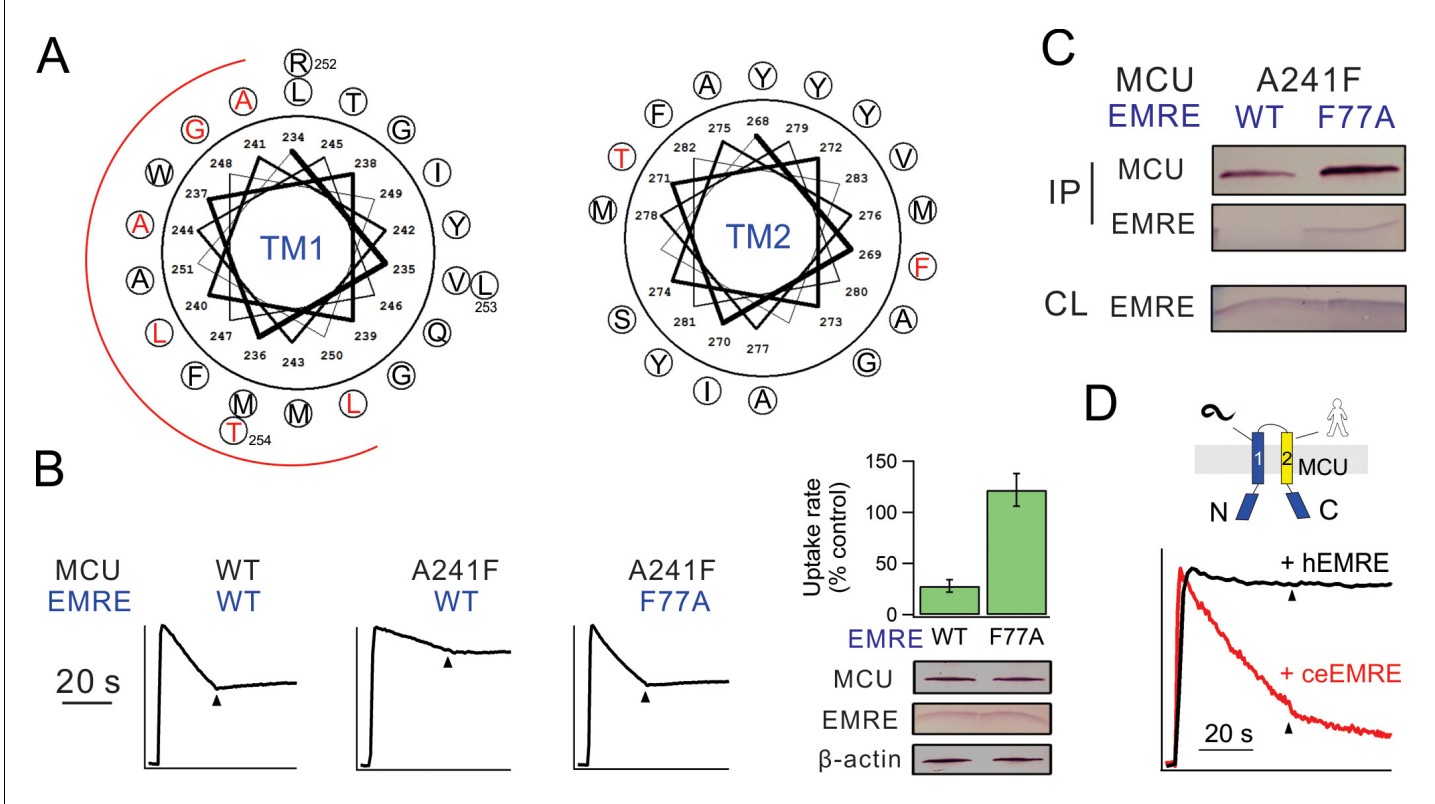

**Figure 4.** Interactions between EMRE and MCU's TMH1. (**A**) Helical projection diagram summarizing functional impact of Trp substitutions in TMHs of MCU. Trp mutation that reduces $Ca^{2+}$ uptake by >70% is defined as high impact (red), and <30% as low impact (black). Red arc highlights proposed helical surface sensitive to Trp substitutions. (**B**) Residue swap showing an impaired MCU mutant (A241F) forming a highly functional uniporter complex with an EMRE mutant (F77A). *Left*: $Ca^{2+}$ uptake in ME-KO cells expressing indicated MCU and EMRE mutants. *Right*: $Ca^{2+}$ uptake (upper) and expression of key uniporter proteins (lower) in cells transfected with A241F-MCU and WT- or F77A-EMRE. (**C**) Co-IP experiments comparing complex formation of A241F-MCU with WT- or F77A-EMRE. (**D**) $Ca^{2+}$ flux in a hMCU-ceMCU chimera (human portion: yellow, *C. elegans* portion: blue), coexpressed with either hEMRE or ceEMRE in ME-KO cells. See also *Figure 4—figure supplements 1–3*.

The following figure supplements are available for figure 4:

**Figure supplement 1.** Functional impact of mutations in MCU's TM helices.

**Figure supplement 2.** Trp mutations in D. discoideum MCU.

**Figure supplement 3.** Uniporter formation by F77A EMRE and MCU mutants.

to test the mutational effect in ME-KO cells. The results (*Figure 4—figure supplement 2*) show that six of these mutations are functionally disruptive while two are fully active; these two active mutants correspond to L240W and A241W in TMH1 of human MCU, suggesting that these residues in human MCU contact EMRE.

A steric clash between a substituted Trp in MCU and a native residue in EMRE could in principle be alleviated by reducing the side-chain volume of that particular EMRE residue. We chose F77 in EMRE to test this idea, as it is on the same helical face as the GxxxS sequence identified above, and since the large Phe residue enables substantial shortening of the side chain. Accordingly, F77A EMRE was coexpressed with each of the 6 functionally defective MCU Trp mutants of TMH1. This EMRE mutant, which forms a functional channel with WT MCU, rescues $Ca^{2+}$ transport with A241W but not with any of the other mutants (*Figure 4—figure supplement 3*). Similarly, the impaired uniporter function induced by A241F in MCU is rescued by F77A in EMRE. Co-IP experiments further show that A241F (or A241W) MCU pulls down F77A but not WT EMRE (*Figure 4C*, *Figure 4—figure supplement 3*). These results demonstrate that the combination of a large and a small side chain (F

or W, A) on EMRE position 77 and MCU position 241 leads to proper transport function regardless of which protein the residues occupy. This 'side chain swap' experiment argues strongly that A241 in MCU's TMH1 is in close proximity to F77 in the TMH of EMRE in the uniporter complex.

While analyzing MCU/EMRE from various species (*Figure 1C*), we noticed that human MCU (hMCU) forms functional $Ca^{2+}$ channels with human EMRE (hEMRE) or with *C. elegans* EMRE (ceEMRE), but *C. elegans* MCU (ceMCU) supports mitochondrial $Ca^{2+}$ uptake only with ceEMRE (*Figure 4D*). These results present an opportunity to test which of ceMCU's two TMHs is responsible for discriminating against hEMRE. Accordingly, we produced two MCU chimeras, with TMH1 or TMH2 of ceMCU substituted by the corresponding region in hMCU. The chimera containing hMCU TMH1, though well expressed, is transport-inactive in the presence of either hEMRE or ceEMRE. The chimera containing hMCU TMH2, however, is activated by ceEMRE, while remaining unresponsive to hEMRE (*Figure 4D*). This result independently supports the proposal that TMH1 of MCU contains the interaction region for EMRE.

## Localization and inner membrane association of MICUs

With the membrane disposition of the MCU-EMRE complex in hand, we now ask how EMRE interacts with MICU proteins (*Sancak et al., 2013*), which contain no apparent transmembrane

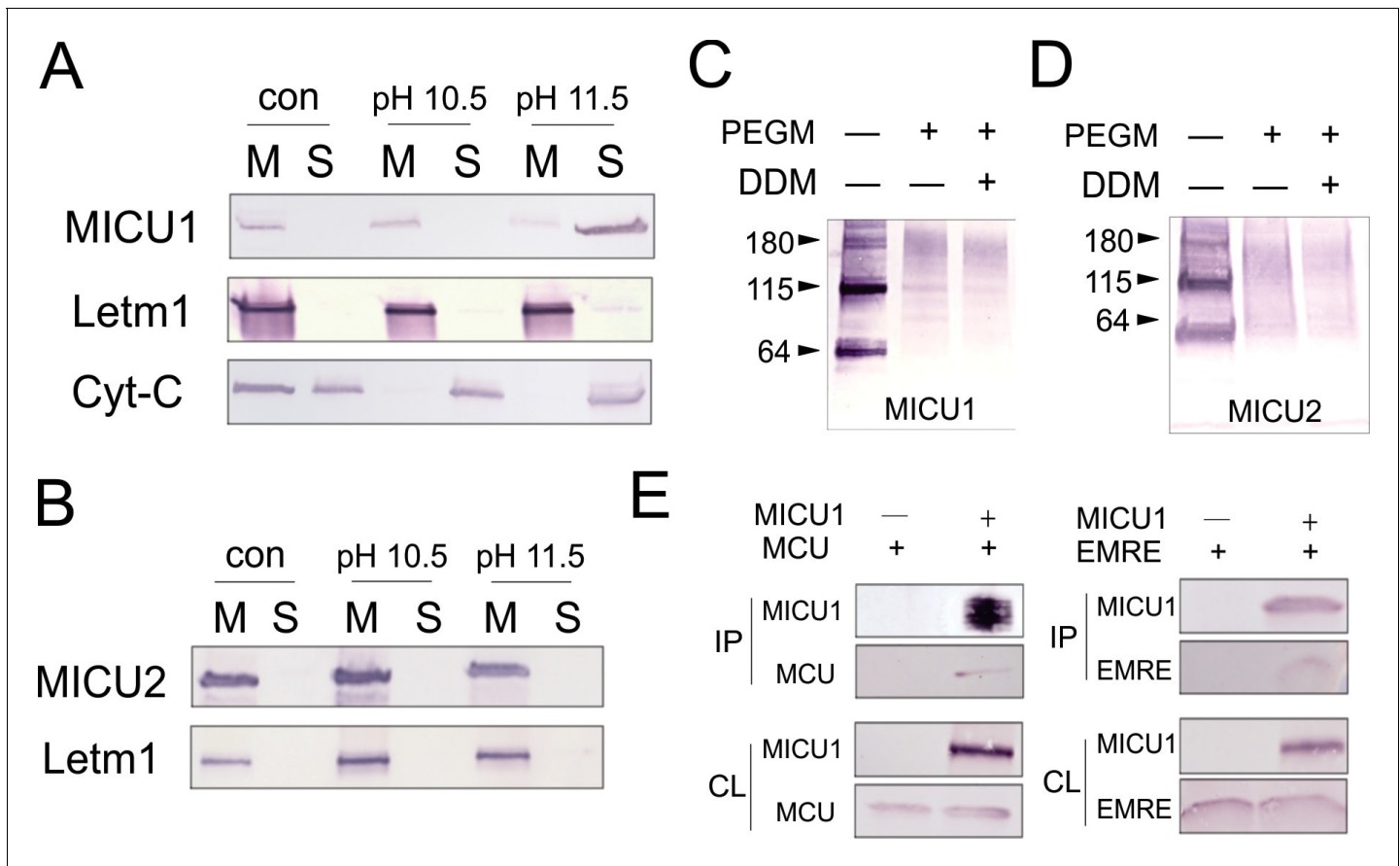

**Figure 5.** Localization of MICUs and interaction with the pore region. (**A–B**) Carbonate extraction (pH 10.5 or 11.5) of MICU1 at 4°C (**A**) or MICU2 at room temperature (**B**) for 1 h, showing membrane pellet (M), proteins extracted into supernatant (S), and control (con) with mitoplasts treated at pH 7.0. (**C– D**) PEGM modification of MICU1 or MICU2. Both MICUs are detected at monomer (~64 kDa) or dimer (~115 kDa) positions. (**E**) Co-IP experiments using immobilized Flag-tagged MICU1 to pull down MCU or EMRE. For all experiments shown in this figure, MICU1 and MICU2 were Flag- and V5-tagged, respectively, and were detected using corresponding Flag and V5 antibodies. See also *Figure 5—figure supplement 1*.

The following figure supplement is available for figure 5:

**Figure supplement 1.** MICU2 interaction with other subunits in the uniporter complex.

sequences. To tackle this problem, we began by testing whether MICUs bind to EMRE from the matrix- or IMS-side of the inner membrane. Currently, the submitochondrial localization of MICU1 is unsettled (*Csordas et al., 2013*; *Foskett and Madesh, 2014*; *Hoffman et al., 2013*; *Hung et al., 2014*; *Waldeck-Weiermair et al., 2015*), and that of MICU2 has not been approached with rigorous methods (*Vais et al., 2016*). Moreover, although MICU1 is known to be a peripheral membrane protein (*Csordas et al., 2013*), it remains unclear if MICU2 is similarly attached to the inner membrane.

Inner membrane association of MICUs was probed by stripping mitoplasts of peripheral membrane proteins using alkaline $Na_2CO_3$ treatments. *Figure 5A* shows that MICU1 and cytochrome C (Cyt-C), but not the integral membrane protein Letm1, are extracted into $Na_2CO_3$ solution, a result consistent with a previous report (*Csordas et al., 2013*) that MICU1 is a peripheral membrane protein. Furthermore, nearly all MICU1 is membrane-bound, virtually none appearing in the IMS without $Na_2CO_3$ treatment, in contrast to Cyt-C, which is found in both the IMS and the membrane (*Figure 5A*). Similar experiments demonstrate that MICU2 remains membrane-associated even after harsher $Na_2CO_3$ extraction conditions (*Figure 5B*). The results thus establish that MICU1 and MICU2 are both confined to the inner-membrane surface under physiological conditions.

To test if MICU1 is exposed to the mitochondrial matrix or the IMS, we incubated mitoplasts with PEGM, relying on MICU1's native cysteines to report accessibility to the reagent from the mitoplast exterior. PEGM readily reacts with MICU1 (*Figure 5C*), showing that this subunit resides on the outer, IMS side of the inner membrane. Two issues regarding this experiment require comment. First, the PEGM-treated sample appears as a smear instead of a defined band, consistent with heterogeneous modification of the protein's 7 native cysteines. Second, a significant western-blot signal is observed roughly corresponding to MICU1 dimers, stable in SDS-PAGE conditions, as reported previously (*Patron et al., 2014*). This signal also shifts upward after PEGM treatment, suggesting that MICU1 dimers or higher oligomers are also localized to the IMS. Similar results in parallel experiments with MICU2 (*Figure 5D*) argue that both MICU1 and MICU2 are associated with the outer leaflet of the mitochondrial inner membrane, a location in harmony with the cytoplasmic $Ca^{2+}$-sensing function of these proteins.

## Interaction of MICUs with the pore-forming region

It has been established, as we also confirm here (*Figure 5—figure supplement 1*), that MICU1 forms a stable complex with MICU2 (*Kamer and Mootha, 2014*; *Patron et al., 2014*), and that MICU1, but not MICU2, is tightly associated with the MCU-EMRE complex (*Kamer and Mootha, 2014*; *Sancak et al., 2013*). It however remains uncertain if MICU1 interacts with MCU or EMRE (*Hoffman et al., 2013*; *Sancak et al., 2013*), a problem addressed using co-IP to examine association of FLAG-tagged MICU1 with MCU or EMRE expressed individually in ME-KO cells. As illustrated in *Figure 5E*, MICU1 can precipitate EMRE without MCU present and MCU without EMRE present. This observation rules out a required MICU1-interacting surface contributed by both MCU and EMRE. It also invites us to search for the molecular determinants mediating MICU1-EMRE interaction in the absence of MCU.

The IMS localization of MICU1 (*Figure 5C*) implies that EMRE binds to MICU1 via its C-terminal, IMS-exposed region containing the polyaspartate tail (EDDDDDD). This highly charged tail alerts us to a complementary polybasic sequence (KKKKR), which though conserved in MICU1, is absent in MICU2 (*Hoffman et al., 2013*). Indeed, co-IP experiments demonstrate that MICU1 pulls down ΔN- but not ΔC-EMRE (*Figure 6A*). Moreover, a MICU1 mutant carrying an electrostatically neutered sequence (KKKKR =>EQEQR) readily complexes with MICU2, but not with EMRE (*Figure 6B*). These results strongly argue that the EMRE-MICU1 interaction is mediated by this electrostatic pair. The strict conservation of these charged sequences further suggests that the EMRE-MICU1 interaction plays an important, previously unappreciated physiological role.

## Functional role of the MICU1-EMRE interaction

The MICU proteins act as the $Ca^{2+}$-sensing gate in the uniporter complex, shutting the pore at resting cellular $Ca^{2+}$ concentrations and opening it when cytoplasmic $Ca^{2+}$ exceeds ~1 µM. By binding to both MCU and MICU1, EMRE might serve as an anchor to retain the MICU1-MICU2 pair near the $Ca^{2+}$-conducting pore. If so, disrupting the MICU1-EMRE interaction would yield a population of channels free of MICUs, allowing unregulated, constitutive $Ca^{2+}$ permeation from the cytosol into

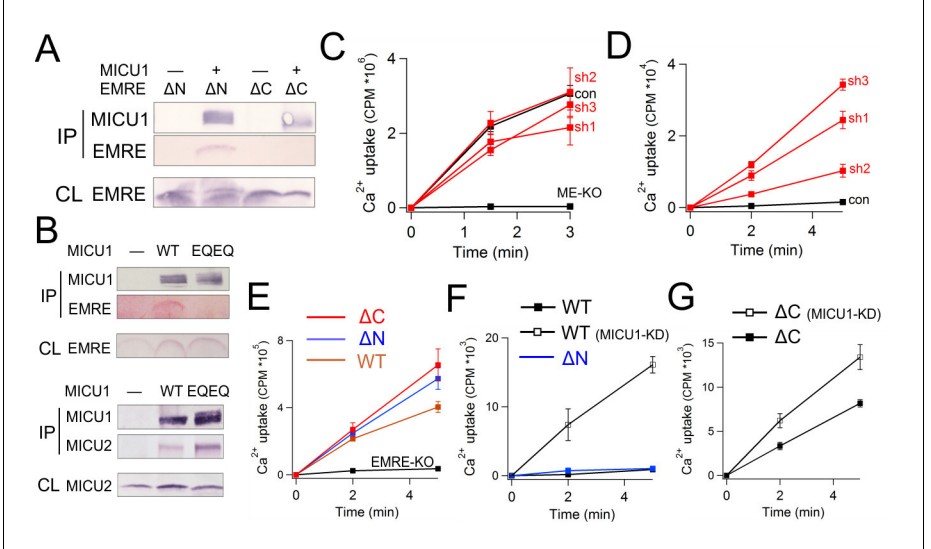

**Figure 6.** Functional importance of the MICU1-EMRE interaction. (**A–B**) Co-IP experiments with WT- or EQEQ-MICU1 (Flag-tagged) used to pull down WT or mutant EMRE proteins in ME-KO cells. MICU1 was detected using anti-Flag, MICU2 by anti-V5, and $\Delta$N- or $\Delta$C-EMRE by anti-C8. (**C–D**) The effect of MICU1 knockdown on mitochondrial $Ca^{2+}$ uptake in WT HEK 293 cells at high (**C**) or low (**D**) $Ca^{2+}$ conditions. Con: cells with no MICU KD. sh1-3 indicates three stable cell lines expressing distinct shRNAs against *MICU1* mRNA. (**E**) Mitochondrial $Ca^{2+}$ uptake (30 μM $Ca^{2+}$) using untransfected EMRE-KO cells, or cells expressing WT-, $\Delta$N-, or $\Delta$C-EMRE as indicated. (**F–G**) $Ca^{2+}$ flux (0.5 μM $Ca^{2+}$) via MCU complexed with WT- or $\Delta$N-EMRE (**F**), or $\Delta$C-EMRE (**G**) in the presence or absence of stable MICU1 KD by shRNA2. Data shown in **C–G** represent mean ± s.e.m. of 3–4 independent measurements. See also *Figure 6—figure supplement 1*.

The following figure supplement is available for figure 6:

**Figure supplement 1.** Biochemical characterization of MICU1 knockdown cells.

the matrix. To examine this idea, we quantify MCU-dependent $Ca^{2+}$ uptake by following accumulation of the $^{45}Ca^{2+}$ radioisotope into mitochondria in digitonin-permeabilized cells, an approach that allows free extramitochondrial $Ca^{2+}$ to be buffered at well-defined submicromolar concentrations without sacrificing sensitivity.

We first examine the effect of MICU1 knockdown (KD) on mitochondrial $Ca^{2+}$ uptake. Three HEK293 cell lines stably expressing distinct short hairpin RNAs (sh1 – 3) were generated, all exhibiting at least 70% decrease of *MICU1* mRNA and normal levels of MCU or EMRE protein (*Figure 6—figure supplement 1*). Although insufficiently sensitive antibodies frustrate quantification of MICU1, the MICU1-KD cell lines show a profound functional alteration in $^{45}Ca^{2+}$ uptake. At high $Ca^{2+}$ (30 μM), rates of $Ca^{2+}$ transport into WT and MICU1-KD mitochondria are similar, while as expected, uptake is virtually undetectable in ME-KO cells (*Figure 6C*). At low $Ca^{2+}$ (0.5 μM), however, MICU1-KD mitochondria show massive $Ca^{2+}$ accumulation in the matrix (*Figure 6D*), in dramatic contrast to WT, where very little $Ca^{2+}$ uptake occurs. These observations confirm previous studies that identified MICU1 as a molecular element controlling $Ca^{2+}$-dependent activation of the uniporter complex (*Csordas et al., 2013*; *Mallilankaraman et al., 2012*).

Functional manifestations of the EMRE-MICU1 interaction were further examined by comparing $^{45}Ca^{2+}$ uptake supported by EMRE variants expressed in EMRE-KO cells. As above, WT-, $\Delta$N-, or $\Delta$C-EMRE all activate MCU-dependent $Ca^{2+}$ uptake to a similar degree at high $Ca^{2+}$ (*Figure 6E*). At low $Ca^{2+}$, uptake is suppressed in mitochondria hosting WT or $\Delta$N-EMRE, but is enhanced over 50-fold by MICU1 knockdown (*Figure 6F*). In contrast, $Ca^{2+}$ rapidly enters mitochondria containing $\Delta$C-EMRE, which cannot bind MICU1, and the rate is only trivially increased after MICU1 KD (*Figure 6G*). We thus conclude that MICU1 must bind EMRE to maintain uninterrupted engagement with the MCU pore, thus conferring $Ca^{2+}$-dependent gating upon what would otherwise be constitutive $Ca^{2+}$ leakage into the mitochondrial matrix.

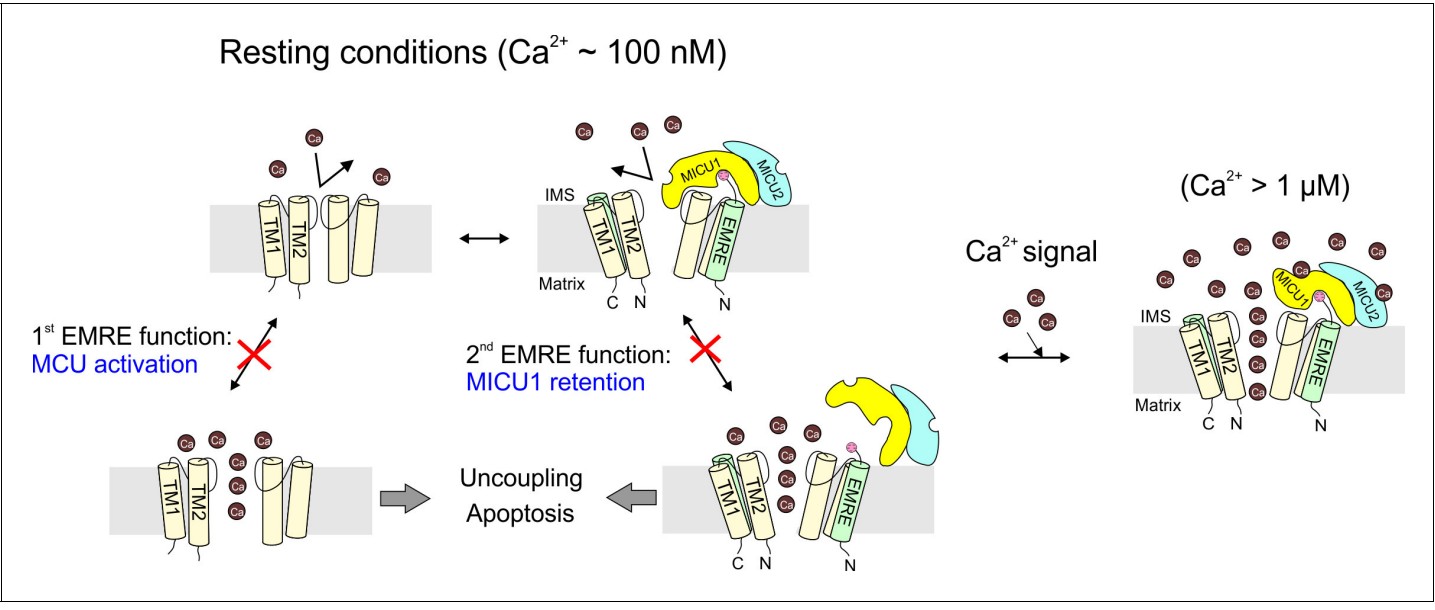

**Figure 7.** The physiological role of EMRE. Cartoon summarizing main findings here, illustrating how the two functions of EMRE, MCU activation and MICU1 retention, together prevent $Ca^{2+}$ leakage through the uniporter complex in resting cellular conditions.

## Discussion

In the past few years, the mitochondrial $Ca^{2+}$ transport field has witnessed a molecular dawn following a half-century of functional phenomenology. It is now firmly established that the human mitochondrial $Ca^{2+}$ uniporter is a $Ca^{2+}$-activated $Ca^{2+}$ channel composed of at least four proteins: MCU, EMRE, MICU1, and MICU2. The present work focuses mainly on EMRE, the least understood component of the channel complex. Impressed by the functional dependence on EMRE arising as metazoan uniporters evolved, we have endeavored to enrich our current view of the uniporter subunits and the physiological purposes of the domain interactions mediating their assembly. Results here establish (1) EMRE orientation in the inner membrane, (2) the molecular contacts governing EMRE interactions with MCU and MICU1, (3) the disposition of the MICU1-MICU2 complex on the outer surface of the inner-membrane, and (4) the functional purpose of the EMRE-MICU1 interaction. These findings lead to a molecular model (*Figure 7*) featuring a central role of EMRE in orchestrating uniporter responses to intracellular $Ca^{2+}$ signaling.

When a $Ca^{2+}$ channel in the plasma membrane opens, or when an intracellular store releases $Ca^{2+}$, a cytoplasmic $Ca^{2+}$ wave is generated. Once the wave hits mitochondria, $Ca^{2+}$ can rise above ~1 µM, activating the uniporter to catalyze rapid $Ca^{2+}$ entry into the mitochondrial matrix. Mitochondria can therefore serve as a buffer to shape intracellular $Ca^{2+}$ signals (*Demaurex et al., 2009*). Moreover, $Ca^{2+}$ entry boosts ATP output by accelerating the citric acid cycle, but excessive, sustained $Ca^{2+}$ accumulation in the matrix triggers caspase-dependent apoptosis (*Rizzuto et al., 2012*). Thus, mitochondria can also decode $Ca^{2+}$ stimulation as either metabolic or death signals. Failure of the uniporter to appropriately respond to $Ca^{2+}$ signals would perturb these crucial physiological processes and could also produce other serious problems. For instance, if the uniporter fails to stay inactive under resting conditions, the large negative inner membrane potential of energized mitochondria would drive continual $Ca^{2+}$ influx. Removing these $Ca^{2+}$ ions requires the action of $Na^+$/$Ca^{2+}$ and $Na^+$/$H^+$ exchangers at a cost of 3 $H^+$ entering into the matrix for each $Ca^{2+}$ expelled. Unregulated, 'leaky' uniporters would therefore divert protons away from the $F_oF_1$-ATPase, partially uncoupling electron transport from ATP synthesis.

The recent discovery of MICU proteins (*Baughman et al., 2011*; *Mallilankaraman et al., 2012*) has enhanced our mechanistic understanding of how cellular $Ca^{2+}$ signals control uniporter activity. It is now clear that the mitochondrial response to physiological $Ca^{2+}$ is mediated by the $Ca^{2+}$-sensing gate formed by the MICU subunits that engage the pore-lining MCU proteins at the external face of

the inner membrane. This modular design comes with a potential danger: that the gating and ion-conducting subunits might dissociate sporadically, producing unregulated $Ca^{2+}$ channels. How could this problem be prevented? It is likely that MICU1 has a high intrinsic affinity to MCU, since at least part of the MICU1-MCU complex survives lengthy co-IP experiments. Moreover, diffusion of MICUs confined to two dimensions within the inner membrane raises the local density of MICUs near the MCU pore. These factors combined could in principle reduce the population of MICU-free, unregulated uniporters — a condition that might be particularly helpful to protist and plant mitochondria, where MCU and MICUs are the only components of the uniporter complex.

The small, single-pass membrane protein EMRE emerges in animals as a new subunit of the uniporter complex. A crucial finding here is that the presence of EMRE in the uniporter complex ensures that the channel conducts $Ca^{2+}$ only when cytoplasmic $Ca^{2+}$ rises above resting levels. This requires EMRE to use its polyaspartate tail to bind MICU1, as if it functions as 'molecular glue' to prevent dissociation of MICU1 from the MCU pore, a circumstance that would produce catastrophic $Ca^{2+}$ leakage (*Figure 7*). Alternatively, EMRE might allosterically transmit the $Ca^{2+}$ signal from MICUs to the pore; in this case, disrupting EMRE-MICU1 interaction would also prevent MICUs from properly gating the $Ca^{2+}$ pathway. We consider this allosteric scenario unlikely, as the mechanism by which MICU1 gates MCU probably evolved in early eukaryotic evolution when EMRE was absent.

The understanding that EMRE safeguards mitochondria against inappropriate $Ca^{2+}$ uptake helps us appreciate the physiological importance of the strict EMRE-dependence of uniporter function appearing in animals. As in any multisubunit protein, it is inevitable that EMRE might occasionally dissociate from MCU, and some tissues under natural or pathological conditions might express MCU in excess of EMRE. Under these situations, a population of EMRE-free uniporters could arise. These channels would also lack MICUs, which would no longer be EMRE-linked to the pore. But the MCU-activation function of EMRE would ensure that these channels would become inactive, preventing them from wreaking havoc on normal cell physiology (*Figure 7*).

We should point out that our results clash with several published assertions regarding the uniporter complex. First, single-channel recordings in planar lipid bilayers have been used to argue that the human MCU protein alone is sufficient to reconstitute a $Ca^{2+}$ channel without EMRE (*De Stefani et al., 2011*; *Patron et al., 2014*). These recordings, however, obtained with in vitro-expressed protein of uncharacterized purity, show channel properties vastly different from uniporter currents directly patch-recorded from intact mitoplasts (*Kirichok et al., 2004*). Second, a recent study (*Vais et al., 2016*) using protease digestion argues that EMRE adopts a $N_{out}$-$C_{in}$ orientation, opposite to that deduced here. Interpretation of the assay, however, is based on an unjustified assumption that the N- and C- termini of EMRE are digested at similar rates. In contrast, our results are supported by two lines of direct and independent evidence – mass tagging of substituted cysteines and the functional competence of the MCU-EMRE fusion protein (*Figure 1*). This same study also claims that EMRE uses its C-terminal tail to sense matrix $Ca^{2+}$, producing a biphasic response of uniporter activity to matrix $Ca^{2+}$. However, this phenomenon, to our best knowledge, has not been observed in any mitochondrial $Ca^{2+}$ uptake experiments in the literature or in previous patch recordings (*Kirichok et al., 2004*). We also note a recent study appearing when our work was under review (*Yamamoto et al., 2016*) that deduced, using an epitope-tagging method, a $N_{in}$-$C_{out}$ EMRE orientation fully consistent with our results. That study also shows that a Pro-to-Ala substitution 3 residues N-terminal to the predicted TMH - a region left unperturbed in our 22-residue deletion ΔN-EMRE construct - abolishes EMRE-MCU interaction. Thus, a small portion in the N-terminus of EMRE near the TMH might also be involved in binding MCU.

In summary, our results argue that EMRE mediates two distinct functions – MCU activation and MICU retention - through two distinct types of subunit-subunit interactions. These functions conspire to achieve a single physiological outcome: obligatory linkage of the $Ca^{2+}$-conducting and $Ca^{2+}$-sensing machineries, a necessary condition for the uniporter complex to respond rapidly and accurately to the elaborate $Ca^{2+}$-signaling network that has evolved in animal cells.

## Materials and methods

### Molecular biology, cell culture, and transient expression

Site-directed mutagenesis was performed using the QuickChange mutagenesis kit (Agilent). HEK 293 cells were grown in Dulbecco's modified Eagle's medium supplemented with 10% FBS, and were incubated at 37°C, 5% $CO_2$. The HEK-293 cell line was supplied by Dr. D.E. Clapham and authenticated by short tandem repeat profiling conducted by ATCC, and was free of mycoplasma as determined by PCR based detection using a kit supplied by ATCC (30-1012K). Transient transfection was performed using Lipofectamine 3000 (Life Technologies), following manufacturer's instructions. Cells were used for downstream analysis 1-2 days after transfection.

### RNA interference

Stable knockdown was achieved by lentivirus, using the transfer vector pLKO.1 puro (Sigma) for U6-driven shRNA expression. The viral titer was determined with a p24 ELISA kit (Clontech, Mountain View, CA). WT HEK293 cells were exposed to the virus for 12 hr, using a multiplicity of infection of 5–10. Afterwards, the culture was incubated with 2 µg/mL puromycin for 2 days to eliminate untransduced cells. The efficiency of knockdown was evaluated by quantitative PCR (qPCR). Detailed qPCR procedure and the shRNA sequences are reported in Extended Experimental Procedures.

### Gene knockout by CRISPR/Cas9

Gene knockout by CRISPR/Cas9 was performed using the published protocol (*Ran et al., 2013*). In brief, the pSpCas9(BB) vector containing the 20-nucleotide guide sequence was transfected into HEK 293 cells. After two days of incubation, single cells were isolated by serial dilutions, and expanded for 2–4 weeks. Gene KO was assessed by sequencing and Western blot. Two sets of guide sequences (see *Supplementary file 1*) were used to rule out off-target effects.

### Western blot and co-immunoprecipitation

For Western blot, proteins on an SDS gel were transferred onto nitrocellulose membranes, which were blocked by 5% milk in TBS, and then incubated with the primary antibody diluted in TBST (TBS + 0.1% Tween-20). Signal development was done using alkaline phosphatase conjugated secondary antibodies (Pierce) and the NBT/BCIP substrate (Life Technologies). The primary antibody and dilution used: α-MCU (Sigma, HPA016480, 1:2000), α-EMRE (Santa Cruz, 86337, 1:400), α-FLAG (Sigma, F1804, 1:4000), α-V5 (Life Technologies, 46–0705, 1:5000), α-Cyt-C (Santa Cruz, 13156, 1:1000), α-β-actin (Santa Cruz, 69879, 1:500), α-Letm1 (Abcam, 55434, 1:2000). Monoclonal anti-1D4 and -C8 antibodies were produced in house.

All co-IP experiments were performed at 4 °C. Transfected HEK 293 cells were grown in a 10-cm dish to confluency, were harvested, and then lysed in 1-mL solubilization buffer (SB, 100 mM NaCl, 20 mM Tris, 1 mM EGTA, 25 mM DDM, pH 7.5-HCl), supplemented with an EDTA-free protease inhibitor cocktail (cOmplete Ultra, Roche). The cell lysate was clarified by centrifugation, and a small portion of the sample was taken for whole cell lysate analysis. Antibody-conjugated Sepharose beads (25 µL) were added, and after 1 h, the beads were collected on a mini column, washed with 2-mL SB, and eluted with 200-µL SDS-gel loading buffer for Western blot. Antibody affinity gel used: FLAG (Sigma, A2220), V5 (Sigma, A7345). 1D4 and C8 affinity gels were produced using 20-mg 1D4 or C8 antibody per 1-g Sepharose 4B (GE Healthcare).

### Mitoplast production

Mitoplasts were formed at 4°C by standard procedures that yield outside-out, stable transport vesicles. Protease inhibitor (cOmplete Ultra, Roche) was present in all steps. HEK 293 cells from a 15-cm dish were pelleted, resuspended in 2-mL mitochondria resuspension buffer (MRB, 250 mM sucrose, 5 mM HEPES, 1 mM EGTA, pH 7.2-KOH), and lysed by passing through a 27.5 g needle 15 – 20 times. Nuclei and cell debris were removed by spinning the cell lysate at 1000 g for 10 min. The supernatant was spun down at 10,000 g for 10 min, resuspended in 2-mL MRB, and then spun down again to pellet crude mitochondria. To obtain mitoplasts, mitochondria were resuspended in 800-µL hypotonic shock buffer (5 mM sucrose, 5 mM HEPES, 1mM EGTA, pH 7.2-KOH), and subjected to osmotic shock for 10 min. Then 200 µL of high-salt storage buffer (750 mM KCl, 100 mM HEPES,

2.5 mM EGTA, pH 7.2-KOH) was added, and mitoplasts were subsequently sedimented by centrifugation at 20,000 g for 10 min. The supernatant, which contains proteins in the outer membrane and the intermembrane space, was collected if further analysis is required.

## Thiol modification

Mitoplasts were resuspended in the thiol-modification buffer (100 mM NaCl, 50 mM MOPS, pH 7.0-NaOH), to which 1 mM PEGM (Sigma) in the presence or absence of 1 mM DDM (Anatrace) was added. The samples were incubated for 1–4 hr at RT before the reaction was quenched with 5 mM cysteine. All reagents were prepared fresh before experiments.

## Carbonate extraction

Mitoplasts were resuspended either in carbonate extraction buffer (120 mM $NaCO_3$, pH 10.5- or 11.5-NaOH) or in a control solution (250 mM sucrose, 25 mM Tris, pH 7.0-HCl). The samples were incubated at 4°C or RT for 1 hr, and then spun down with ultracentrifugation at 200,000 g for 1 hr. The supernatant contains proteins extracted by carbonate, while the membrane pellet containing integral membrane proteins.

## Mitochondrial $Ca^{2+}$ uptake assays

All $Ca^{2+}$ uptake assays were repeated at least 3 times on multiple preparations, and traces in figures show typical responses. For the fluorescence-based assay, $10^7$ HEK 293 cells were suspended in 10-mL $Ca^{2+}$ flux wash buffer (CWB, 120 mM KCl, 25 mM HEPES, 2 mM $KH_2PO_4$, 1 mM $MgCl_2$, 50 μM EGTA, pH 7.2-KOH), pelleted at 1000 g for 5 min, and resuspended in 2.5-mL recording buffer (RB, 120 mM KCl, 25 mM HEPES, 2 mM $KH_2PO_4$, 1 mM $MgCl_2$, 5 mM succinate, pH 7.2-KOH). 2 mL of the cell suspension was loaded into a stirred quartz cuvette in a Hitachi F-2500 spectrophotometer (ex: 506 nm, ex-slit: 2.5 nm, em: 532 nm, em-slit: 2.5 nm, sampling frequency: 2 Hz), with the temperature maintained at 37°C by a circulating bath. In a typical experiment, reagents were added into the cuvette in the following order: 0.5-μM calcium green 5N (Life Technologies), 30-μM digitonin (Sigma), 10-μM $CaCl_2$, and 2-μM Ru360 (Santa Cruz). Under these conditions, peak free $Ca^{2+}$ concentrations were close to 10 μM (11 $\pm$ 6 SD, N=40, as determined by calibration). Because of uncertainties in protein concentration, $Ca^{2+}$ uptake activity is reported as a linear fit to the fluorescent signal obtained in the first 10 s after addition of 10-μM $CaCl_2$. Activity was not altered by 5 μM thapsigargin.

For the $^{45}Ca^{2+}$ based uptake assay, 2 x $10^6$ cells were suspended in 1.5-mL CWB, spun down at 2,000 g for 1 min, and resuspended again in 200-μL CWB, supplemented with 5 μM thapsigargin and 30 μM digitonin. To initiate $Ca^{2+}$ flux, 100 μL of the cell suspension was transferred to either 400-μL high-$Ca^{2+}$ flux buffer (RB + 10 μM EGTA and 40 μM $^{45}CaCl_2$) or 400-μL low-$Ca^{2+}$ flux buffer (RB + 0.69 mM EGTA, 0.5 mM $CaCl_2$, and 20 uM $^{45}CaCl_2$, pH 7.2-KOH), with both solution containing 5-μM thapsigargin and 30-μM digitonin. At desired time points, 100 μL of the reaction mixture was added into 5-mL ice-cold CWB, and then filtered through 0.45-μm nitrocellulose membranes (EMD-Millipore) on a vacuum filtration manifold (Millipore model 1225). The membrane was washed immediately with 5-mL ice cold CWB, and later transferred into scintillation vials for counting. $^{45}Ca^{2+}$ radioisotope was purchased from Perkin Elmer, with a specific activity of 12.5 mCi/mg.

## Bioinformatics

Sequences of MCU or EMRE homologues were collected using PSI-BLAST search of ~100 species. EMRE was identified by the presence of the polyaspartic tail, and MCU by the conserved DIME loop. Multiple sequence alignment was performed using the ClustalW2 online server (*Larkin et al., 2007*). The helical wheels were plotted using Antheprot v 6.4 (*Deleage et al., 2001*). Mitochondrial targeting sequence prediction was carried out using the TargetP 1.1 online server (*Emanuelsson et al., 2007*).

## Quantitative PCR

Whole cell RNA was extracted from HEK 293 cells grown in 6-well plates using TRIzol. Residual DNA was removed using the TURBO DNA-free kit (Life Technologies). First strand cDNA synthesis was performed with 1 μg RNA using M-MuLV reverse transcriptase (NEB), following manufacturer's

instructions. The sample was subsequently digested with RNaseH (NEB). qPCR was performed with SsoFast EvaGreen Supermixes (BIO-RAD), using 0.5 μM $\beta$-actin or MICU1 primers, and 0.5, 2.5, 5, or 10 ng RNA for producing a standard curve. Detection of the PCR product was done with a CFX96 real-time PCR detection system (BIO-RAD), using the following protocol: 95°C for 30 s, 50 cycles of 95°C for 5 s and 57°C for 5 s. The sequence of the primers is provided in *Supplementary file 1*.

$\Delta$Ct was calculated by subtracting the Ct for $\beta$-actin from the Ct for MICU1, with 3 independent RNA extractions and qPCR measurements using 2.5 ng whole RNA. $\Delta\Delta$Ct was calculated by subtracting the mean $\Delta$Ct for control WT cells from the mean $\Delta$Ct for each stable MICU1 knockdown cells. The results were presented as the percentage of MICU1 mRNA in MICU1 knockdown cells relative to MICU1 mRNA in WT control, using the equation% mRNA = $1/2^{|\Delta\Delta Ct|}$.

## Acknowledgements

We thank Dr. Vamsi Mootha for plasmids and valuable advice, Daniel Turman, Dr. Nicolas Last, Dr. Joel Meyerson, Dr. Randy Stockbridge, and Dr. Rocio Finol-Urdaneta for insightful comments of the manuscript. We are particularly grateful to Dr. Daniel Oprian for assistance for raising antibodies and producing the associated affinity media, and for supporting M.R. to participate in this project. This work is supported in part by NIH Grant R01-GM107023. The authors declare no conflict of interest.

## Additional information

### Funding

| Funder | Grant reference number | Author |
| --- | --- | --- |
| Howard Hughes Medical Institute | | Ming-Feng Tsai |
| National Institutes of Health | R01-GM107023 | Christopher Miller |

The funders had no role in study design, data collection and interpretation, or the decision to submit the work for publication.

### Author contributions

M-FT, Investigation, Writing - Original draft, Supervision, Conception and design, Acquisition of data, Analysis and interpretation of data, Drafting or revising the article; CBP, MR, YW, Investigation, Conception and design, Acquisition of data, Analysis and interpretation of data; C-WT, Investigation, Conception and design, Acquisition of data, Analysis and interpretation of data, Drafting or revising the article; CW, Investigation, Acquisition of data, Analysis and interpretation of data; CM, Funding acquisition, Supervision, Conception and design, Analysis and interpretation of data, Drafting or revising the article

### Author ORCIDs

Christopher Miller, http://orcid.org/0000-0002-0273-8653

## Additional files

**Supplementary files**
• Supplementary file 1. Supplementary experimental procedure.

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
