## [Decision Letter]

Thank you for submitting your article "A Central Role of EMRE in Regulation of the Mitochondrial Uniporter Complex" for consideration by *eLife*. Your article has been favorably evaluated by Gary Westbrook (Senior Editor) and three reviewers, one of whom, David Clapham, is a member of our Board of Reviewing Editors. The reviewers discussed the reviews with one another and the Senior Editor has drafted this decision to help you prepare a revised submission.

As you will see, all reviewers were enthusiastic about the quality and relevance of the work, and its publication after revision in *eLife*. Reviewers 2 and 3 had very minor points as listed below. Reviewer 2 made some extensive comments, but note these were considered "suggestions". We ask that you consider these points in your revised manuscript.

Minor points from Reviewers 2 and 3:

1) A new paper in BBA reports functional mutagenesis of EMRE, showing that deletion of amino acids at EMRE's N terminus abolishes function, in conflict with the current paper. The discrepancy ought to be reconciled or at least discussed.

2) Figure 2: Western blot: what was the molecular weight for EMRE?

3) Figure 5: The predicted mol. wt. for MICU1 and MICU2 are approximately 54 and 50 KDa respectively. The observed bands in Western blots are near 64 KDa for both. How do you explain the difference? You also mentioned that MICU1 antibody was not good enough to quantify the MICU1 expression level in MICU1 knockdown cell lines. A comment on reliability of your MICU1 antibodies may be needed.

4) Figure 5 indicates that MICU2 remains membrane-bound even after harsher Na2CO3 treatment where as MICU1 comes off. This result seems to contradict the reports indicating that MICU2 binds MCU-complex via MICU1 either by forming a disulfide bond or other (Patron et al, 2014 as well as Kamer et al, 2014). This result was not discussed at all considering its importance.

5) Figure 5 shows that MICU1 can precipitate MCU in the absence of EMRE. Doesn't this observation contradict the strict requirement of EMRE C-terminal for MICU1 engagement with the pore-forming subunit (MCU)?

Reviewer #1 general assessment:

Tsai et al. investigate EMRE, a non- pore-forming subunit of the mitochondrial uniporter, (the MCU, or MiCa channel). They approach the relevant orientation and functional questions with a large number of experimental methods. First, the authors use gene editing to develop assays in permeabilized HEK cells. MCU homologs from amoeba and plants mediate Ca uptake alone, while worms, flies, and human MCU requires EMRE. Cysteine and tryptophan scanning/substitution mutagenesis then enables them to verify that MCU's N and C termini are inside the mitoplast, while EMRE's N terminus is inside the mitoplast and its C terminus outside. Chimeras and other experiments then suggest that TMH1 of MCU is the interaction region for EMRE. Other subunits, the relatively soluble subunits MICU1 and MICU2, are shown to associate with the outer leaflet of the inner mitochondrial membrane, and that EMRE associates with MICU1 by electrostatic interactions. MICU function is verified as regulating Ca entry, and the authors propose that MICU1 must bind EMRE to engage the MCU pore. Thus, MICUs regulate what would otherwise be an unregulated Ca pore. An important finding is that EMRE polyaspartate tail ensures that the channel conducts Ca only when cytoplasmic Ca rises above its resting levels. In short, EMRE mediates MICU retention and MCU retention. Based on the lack of EMRE in lower species, the authors conclude that EMRE arose as a safety mechanism, allowing MICU1 gating of MCU only if EMRE is present.

This work clarifies many of the issues that have plagued MCU subunit function and orientation. The methods are robust and the conclusions are likely sound. The only additional experiments the field needs to settle remaining conformational questions is the high resolution structure of the MCU/EMRE/MICU1/2/MCUR1 complex.

Reviewer #2 general assessment and major comments:

This is a new paper from a premiere ion channel biochemistry lab. The paper focuses on the macromolecular organization of the mitochondrial calcium uniporter complex, which in humans consists of five proteins: MCU (the pore), MICU1/2 (calcium sensors in the IMS), and EMRE (a small membrane protein that relays MICU1/2 calcium sensing to the open state of MCU). The study combines clever biochemical methods (e.g., artificial transmembrane domains, tryptophan mutagenesis, and protein fusions) to shed insight into the complex, with a focus on EMRE and its interactions with MCU and MICU1. The topic is timely. Some important suggestions designed to help the authors improve what could be a very important paper:

1) This paper reports a mix of novel and not so novel results – hence novel findings are buried. I would encourage the authors to emphasize the novel and de-emphasize results that are confirmatory. For example, the topology of MCU, IMS localization of MICU1/2, gatekeeping activity by MICU1, and dual functions of EMRE (keeping MCU open and linking MICU1/2 to MCU) are all well-established now. The novel findings include the topology of EMRE, the MCU-EMRE fusion (which could be a very important tool for the field), EMRE interaction with TM1, and interaction between MICU1 and EMRE.

2) In MCU knockout cell lines and mouse tissues, loss of MCU causes loss of EMRE expression. In many figures, the authors claim loss of interaction (by IP) between EMRE and MCU, but EMRE protein expression seems normal. This raises the possibility that the IP conditions used here do not reflect the in vivo binding status of these two proteins. For example, in Figure 3, the authors mutate EMRE and evaluate which variants are able to interact with MCU. The S85W mutant evidently expresses very well, yet does not interact with MCU. Can the authors provide additional evidence, e.g., BN-PAGE, demonstrating a lack of a shift of the MCU holocomplex?

3) The authors claim that the δ-N, δ-C and WALP EMRE localize to the mitochondria. However, it's not clear how pure their mitochondrial preps are without markers for other organelles. Alternatively, they could consider simple microscopy experiments to look for co-localization with mitochondrial markers. Again, it is surprising to see that WALP EMRE expresses so well, even though it does not bind to MCU. How do the activities of δ-N and δ-C compare to WT? Please add WT EMRE traces and expression to Figure 3.

4) One of the important new claims relates to the nature of the interaction between MCU and EMRE. The authors have performed tryptophan scanning mutagenesis of both transmembranes of MCU and identify six in TM1 versus two in TM2 that disrupt function. Based on this difference (six versus two), the authors prioritize TM1 as being more important for EMRE binding. They then show that two of these mutations can be introduced into the DdMCU without impairing function – and since DdMCU operates independently of EMRE, prioritize these two as important for human MCU binding to EMRE. Finally, the authors perform an amino acid swap between MCU and EMRE to support the conclusion that A241 in MCU and F77 in EMRE directly interact.

I have some concerns about the logic and inference. First, it's not clear how the authors can exclude the possibility that the other four residues on TM1, or the other two residues on TM2, are important for EMRE binding. Second, if this model is correct, then the MCU chimera with TM1 from human, and TM2 from *C. elegans*, ought in principle be activated by human EMRE, yet it is not. Third, in these functional studies shown in Figure 3—figure supplement 1, IPs are not performed, so we do not know if EMRE is actually bound or not bound to MCU. The authors ought to review these data, add the necessary controls, and then state their conclusions carefully. Also, the authors cannot claim direct interaction – this is nuanced in the text, but in the figure legends, direct interaction is claimed.

5) The co-IP data presented in Figure 5 suggests that MCU can interact with MICU1 independent of EMRE, and that MICU1 can interact with EMRE independent of MCU. These experiments lack proper controls. The authors could express another mitochondrial IM protein as a negative control.

Reviewer #3 general assessment:

This is an extremely important paper that clarifies the general architecture of the MCU complex and sheds light on the physiological function of EMRE and MICU1. The experiments are carefully planned and conducted. The paper is essentially ready for publication.

---

## [Author Response]

Reviewer #2 major comments:

[…] Some important suggestions designed to help the authors improve what could be a very important paper:

1) This paper reports a mix of novel and not so novel results – hence novel findings are buried. I would encourage the authors to emphasize the novel and de-emphasize results that are confirmatory. For example, the topology of MCU, IMS localization of MICU1/2, gatekeeping activity by MICU1, and dual functions of EMRE (keeping MCU open and linking MICU1/2 to MCU) are all well-established now. The novel findings include the topology of EMRE, the MCU-EMRE fusion (which could be a very important tool for the field), EMRE interaction with TM1, and interaction between MICU1 and EMRE.

We agree that the topology of MCU, IMS localization of MICU1/2, and the gatekeeping function of MICU1 have been established, and we have indeed cited appropriate references in the manuscript. However, we disagree with the reviewer that the dual function of EMRE is established. Prior to our work, the only known function of EMRE is keeping MCU open, but the physiological significance of such function is unclear. To be sure, it has been reported that EMRE interacts with MICU1, but the purpose and mechanism of this interaction, and how this molecular contact regulates uniporter function, have remained unknown until this study. We have endeavored in the Abstract, the Introduction, and the Discussion to emphasize and highlight the major contribution of this work: to provide a unifying picture of how the MCU-activation and MICU1-rentention functions of EMRE together prevent Ca^2+^ leakage through the uniporter during resting cell conditions.

2) In MCU knockout cell lines and mouse tissues, loss of MCU causes loss of EMRE expression. In many figures, the authors claim loss of interaction (by IP) between EMRE and MCU, but EMRE protein expression seems normal. This raises the possibility that the IP conditions used here do not reflect the in vivo binding status of these two proteins. For example, in Figure 3, the authors mutate EMRE and evaluate which variants are able to interact with MCU. The S85W mutant evidently expresses very well, yet does not interact with MCU. Can the authors provide additional evidence, e.g., BN-PAGE, demonstrating a lack of a shift of the MCU holocomplex?

There is so far no evidence that the steady-state expression level of EMRE directly reflects the integrity of the MCU-EMRE interaction. Protein expression is a complicated function of the rate of proteolysis, polypeptide synthesis, protein insertion into membranes, etc., a subject that we have currently under study, and one far beyond the scope of this manuscript. An EMRE mutation such as S85W can in principle influence any of these factors and can thus alter EMRE’s observed expression level.

We have in fact done proper controls for the CoIP experiment, showing that MCU pulls down WT and control Trp mutants, but not G81W and S85W EMRE. BN-PAGE would not provide additional information, since as in CoIP, it requires the protein complex to be first extracted into detergents before being analyzed by Western blot.

3) The authors claim that the δ-N, δ-C and WALP EMRE localize to the mitochondria. However, it's not clear how pure their mitochondrial preps are without markers for other organelles. Alternatively, they could consider simple microscopy experiments to look for co-localization with mitochondrial markers. Again, it is surprising to see that WALP EMRE expresses so well, even though it does not bind to MCU. How do the activities of δ-N and δ-C compare to WT? Please add WT EMRE traces and expression to Figure 3.

In Figure 3, we show that ΔN and ΔC EMRE, but not WALP, support MCU function. There is certainly a low possibility that WALP EMRE fails to activate MCU because it is targeted to other organelles, despite its mitochondrial targeting sequence. This possibility could be ruled out by extensive control experiments, such as the co-localization experiment suggested by the reviewer. However, we cannot justify the time required for these experiments, as at the end of the day it is the extensive Trp mutagenesis screening (Figure 3) and the side-chain swap experiments (Figure 4) that powerfully establish that TMH interactions between MCU and EMRE are necessary for uniporter function.

The activity of ΔN, ΔC, and WT EMRE are similar, as is compared in Figure 6 using a more sensitive ^45^Ca^2+^ flux assay. Again, there is no evidence in the literature that EMRE steady-state expression is controlled solely by MCU binding.

4) One of the important new claims relates to the nature of the interaction between MCU and EMRE. The authors have performed tryptophan scanning mutagenesis of both transmembranes of MCU and identify six in TM1 versus two in TM2 that disrupt function. Based on this difference (six versus two), the authors prioritize TM1 as being more important for EMRE binding. They then show that two of these mutations can be introduced into the DdMCU without impairing function – and since DdMCU operates independently of EMRE, prioritize these two as important for human MCU binding to EMRE. Finally, the authors perform an amino acid swap between MCU and EMRE to support the conclusion that A241 in MCU and F77 in EMRE directly interact.

I have some concerns about the logic and inference. First, it's not clear how the authors can exclude the possibility that the other four residues on TM1, or the other two residues on TM2, are important for EMRE binding. Second, if this model is correct, then the MCU chimera with TM1 from human, and TM2 from C. elegans, ought in principle be activated by human EMRE, yet it is not. Third, in these functional studies shown in Figure 3—figure supplement 1, IPs are not performed, so we do not know if EMRE is actually bound or not bound to MCU. The authors ought to review these data, add the necessary controls, and then state their conclusions carefully. Also, the authors cannot claim direct interaction – this is nuanced in the text, but in the figure legends, direct interaction is claimed.

Our side-chain swap experiment allows us to identify A241 as a residue interacting with EMRE, but we certainly do not exclude the possibility that other residues sensitive to Trp substitution might also be important for EMRE binding. The reviewer is right that our model would predict that an MCU chimera made of human-MCU TM1 and *C. elegans* MCU TM2 should respond to human EMRE. Unfortunately, such chimera fails to respond to both human and *C. elegans* EMRE. We cannot interpret a negative result like this, as the failure to detect function could result from many potential issues, such as misfolding of the chimera protein.

5) The co-IP data presented in Figure 5 suggests that MCU can interact with MICU1 independent of EMRE, and that MICU1 can interact with EMRE independent of MCU. These experiments lack proper controls. The authors could express another mitochondrial IM protein as a negative control.

The reviewer, citing precedents in the literature, suggests a common way to rule out non-specific binding of MICU1 to the affinity column: to show that another mitochondrial IM protein fails to bind to the column. However, we consider this kind of negative control to be a bit indirect. Instead, we opted for a more laborious but more powerful negative control that directly demonstrates that MICU1 is not retained in the column in the absence of MCU or EMRE (Figure 5).

Minor points from Reviewers 2 and 3:

1) A new paper in BBA reports functional mutagenesis of EMRE, showing that deletion of amino acids at EMRE's N terminus abolishes function, in conflict with the current paper. The discrepancy ought to be reconciled or at least discussed.

Yes, we saw this new paper when it appeared. It fully confirms our orientation results. There is in fact no discrepancy, as their substitutions and short deletion at the N-terminus were different from our more extensive deletion. We comment on these new results in the Discussion.

2) Figure 2: Western blot: what was the molecular weight for EMRE?

It is ~10 kDa. We have added this information to the legends.

3) Figure 5: The predicted mol. wt. for MICU1 and MICU2 are approximately 54 and 50 KDa respectively. The observed bands in Western blots are near 64 KDa for both. How do you explain the difference? You also mentioned that MICU1 antibody was not good enough to quantify the MICU1 expression level in MICU1 knockdown cell lines. A comment on reliability of your MICU1 antibodies may be needed.

We do not know why the MICUs travel more slowly than expected. Deviation from the expected ICU1 immunodetection: this is reliable, since MICU1 tagged with FLAG, and detected by anti-FLAG antibody. We have now revised the methods section to make sure that this is clear.

4) Figure 5 indicates that MICU2 remains membrane-bound even after harsher Na2CO3 treatment where as MICU1 comes off. This result seems to contradict the reports indicating that MICU2 binds MCU-complex via MICU1 either by forming a disulfide bond or other (Patron et al, 2014 as well as Kamer et al, 2014). This result was not discussed at all considering its importance.

In our Na_2_CO_3_ extraction experiments, either MICU1 or MICU2 is overexpressed so that one of them is in large molecular excess than the other. This allows us to test interaction of individual MICU with lipid, with minimized complication from MICU1-MICU2 interaction. Our results agree with the idea that MICU1 and MICU2 form a heterodimer, as in the Patron and Kamer paper.

5) Figure 5 shows that MICU1 can precipitate MCU in the absence of EMRE. Doesn't this observation contradict the strict requirement of EMRE C-terminal for MICU1 engagement with the pore-forming subunit (MCU)?

This is a good point but not a fatal one for our argument. Our results show that EMRE C-terminus is necessary for MCU to be “fully” gated by MICU1, but we did not rule out that without this interaction, there could be some MCU-MICU1 interaction that allows some binding and perhaps a less perfectly coupled form of uniporter gating. We actually think, as in Discussion, that this MCU-MICU1 interaction might be crucial for protist and plant uniporters, which do not have EMRE as a subunit.